# Hash Embeddings for Efficient Word Representations

**Dan Svenstrup**
Department for Applied Mathematics and Computer Science
Technical University of Denmark (DTU)
2800 Lyngby, Denmark
dsve@dtu.dk

**Jonas Meinertz Hansen**
FindZebra
Copenhagen, Denmark
jonas@findzebra.com

**Ole Winther**
Department for Applied Mathematics and Computer Science
Technical University of Denmark (DTU)
2800 Lyngby, Denmark
olwi@dtu.dk

## Abstract

We present hash embeddings, an efficient method for representing words in a continuous vector form. A hash embedding may be seen as an interpolation between a standard word embedding and a word embedding created using a random hash function (the hashing trick). In hash embeddings each token is represented by $k$ $d$-dimensional embeddings vectors and one $k$ dimensional weight vector. The final $d$ dimensional representation of the token is the product of the two. Rather than fitting the embedding vectors for each token these are selected by the hashing trick from a shared pool of $B$ embedding vectors. Our experiments show that hash embeddings can easily deal with huge vocabularies consisting of millions of tokens. When using a hash embedding there is no need to create a dictionary before training nor to perform any kind of vocabulary pruning after training. We show that models trained using hash embeddings exhibit at least the same level of performance as models trained using regular embeddings across a wide range of tasks. Furthermore, the number of parameters needed by such an embedding is only a fraction of what is required by a regular embedding. Since standard embeddings and embeddings constructed using the hashing trick are actually just special cases of a hash embedding, hash embeddings can be considered an extension and improvement over the existing regular embedding types.

## 1 Introduction

Contemporary neural networks rely on loss functions that are continuous in the model's parameters in order to be able to compute gradients for training. For this reason, any data that we wish to feed through the network, even data that is of a discrete nature in its original form will be translated into a continuous form. For textual input it often makes sense to represent each distinct word or phrase with a dense real-valued vector in $\mathbb{R}^n$. These word vectors are trained either jointly with the rest of the model, or pre-trained on a large corpus beforehand.

For large datasets the size of the vocabulary can easily be in the order of hundreds of thousands, adding millions or even billions of parameters to the model. This problem can be especially severe when $n$-grams are allowed as tokens in the vocabulary. For example, the pre-trained Word2Vec vectors from Google (Miháltz, 2016) has a vocabulary consisting of 3 million words and phrases. This means that even though the embedding size is moderately small (300 dimensions), the total number of parameters is close to one billion.

The embedding size problem caused by a large vocabulary can be solved in several ways. Each of the methods have some advantages and some drawbacks:

1. **Ignore infrequent words**. In many cases, the majority of a text is made up of a small subset of the vocabulary, and most words will only appear very few times (Zipf's law (Manning et al., 1999)).

   By ignoring anything but most frequent words, and sometimes stop words as well, it is possible to preserve most of the text while drastically reducing the number of embedding vectors and parameters. However, for any given task, there is a risk of removing too much or to little. Many frequent words (besides stop words) are unimportant and sometimes even stop words can be of value for a particular task (e.g. a typical stop word such as "and" when training a model on a corpus of texts about logic). Conversely, for some problems (e.g. specialized domains such as medical search) rare words might be very important.

2. **Remove non-discriminative tokens after training**. For some models it is possible to perform efficient feature pruning based on e.g. entropy (Stolcke, 2000) or by only retaining the $K$ tokens with highest norm (Joulin et al., 2016a). This reduction in vocabulary size can lead to a decrease in performance, but in some cases it actually avoids some over-fitting and increases performance (Stolcke, 2000). For many models, however, such pruning is not possible (e.g. for on-line training algorithms).

3. **Compress the embedding vectors**. Lossy compression techniques can be employed to reduce the amount of memory needed to store embedding vectors. One such method is quantization, where each vector is replaced by an approximation which is constructed as a sum of vectors from a previously determined set of centroids (Joulin et al., 2016a; Jegou et al., 2011; Gray and Neuhoff, 1998).

For some problems, such as online learning, the need for creating a dictionary before training can be a nuisance. This is often solved with *feature hashing*, where a hash function is used to assign each token $w \in \mathcal{T}$ to one of a fixed set of "buckets" $\{1, 2, \ldots B\}$, each of which has its own embedding vector. Since the goal of hashing is to reduce the dimensionality of the token space $\mathcal{T}$, we normally have that $B \ll |\mathcal{T}|$. This results in many tokens "colliding" with each other because they are assigned to the same bucket. When multiple tokens collide, they will get the same vector representation which prevents the model from distinguishing between the tokens. Even though some information is lost when tokens collide, the method often works surprisingly well in practice (Weinberger et al., 2009).

One obvious improvement to the feature hashing method described above would be to learn an optimal hash function where important tokens do not collide. However, since a hash function has a discrete codomain, it is not easy to optimize using e.g. gradient based methods used for training neural networks (Kulis and Darrell, 2009).

The method proposed in this article is an extension of feature hashing where we use $k$ hash functions instead of a single hash function, and then use $k$ trainable parameters for each word in order to choose the "best" hash function for the tokens (or actually the best combination of hash functions). We call the resulting embedding *hash embedding*. As we explain in section 3, embeddings constructed by both feature hashing and standard embeddings can be considered special cases of hash embeddings.

A hash embedding is an efficient hybrid between a standard embedding and an embedding created using feature hashing, i.e. a hash embedding has all of the advantages of the methods described above, but none of the disadvantages:

- When using hash embeddings there is no need for creating a dictionary beforehand and the method can handle a dynamically expanding vocabulary.
- A hash embedding has a mechanism capable of implicit vocabulary pruning.
- Hash embeddings are based on hashing but has a trainable mechanism that can handle problematic collisions.
- Hash embeddings perform something similar to product quantization. But instead of all of the tokens sharing a single small codebook, each token has access to a few elements in a very large codebook.

Using a hash embedding typically results in a reduction of parameters of several orders of magnitude. Since the bulk of the model parameters often resides in the embedding layer, this reduction of

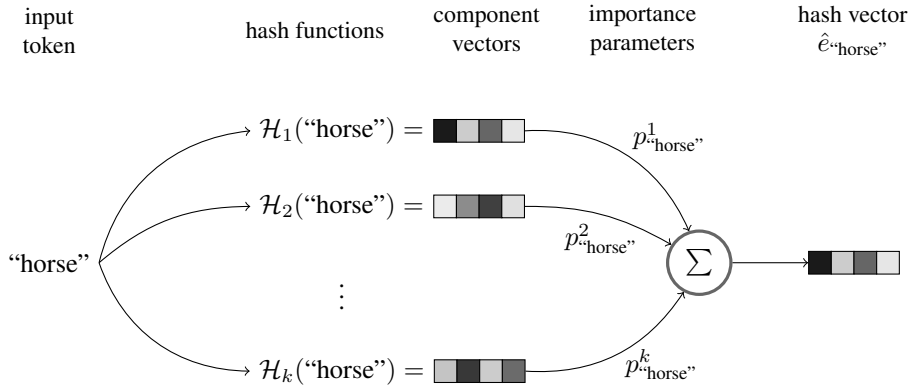

Figure 1: Illustration of how to build the hash vector for the word "horse". The optional step of concatenating the vector of importance parameters to $\hat{e}_{\text{"horse"}}$ has been omitted. The size of component vectors in the illustration is $d = 4$.

parameters opens up for e.g. a wider use of e.g. ensemble methods or large dimensionality of word vectors.

## 2   Related Work

Argerich et al. (2016) proposed a type of embedding that is based on hashing and word co-occurrence and demonstrates that correlations between those embedding vectors correspond to the subjective judgement of word similarity by humans. Ultimately, it is a clever reduction in the embedding sizes of word co-occurrence based embeddings.

Reisinger and Mooney (2010) and since then Huang et al. (2012) have used multiple different word embeddings (prototypes) for the same words for representing different possible meanings of the same words. Conversely, Bai et al. (2009) have experimented with hashing and treating words that co-occur frequently as the same feature in order to reduce dimensionality.

Huang et al. (2013) have used bags of either bi-grams or tri-grams of letters of input words to create feature vectors that are somewhat robust to new words and minor spelling differences.

Another approach employed by Zhang et al. (2015); Xiao and Cho (2016); Conneau et al. (2016) is to use inputs that represent sub-word units such as syllables or individual characters rather than words. This generally moves the task of finding meaningful representations of the text from the input embeddings into the model itself and increases the computational cost of running the models (Johnson and Zhang, 2016). Johansen et al. (2016) used a hierarchical encoding technique to do machine translation with character inputs while keeping computational costs low.

## 3   Hash Embeddings

In the following we will go through the step by step construction of a vector representation for a token $w \in \mathcal{T}$ using hash embeddings. The following steps are also illustrated in fig. 1:

1. Use $k$ different functions $\mathcal{H}_1, \ldots, \mathcal{H}_k$ to choose $k$ *component vectors* for the token $w$ from a predefined pool of $B$ shared component vectors

2. Combine the chosen component vectors from step 1 as a weighted sum: $\hat{e}_w = \sum_{i=1}^{k} p_w^i \mathcal{H}_i(w)$. $p_w = (p_w^1, \ldots, p_w^k)^\top \in \mathbb{R}^k$ are called the *importance parameters* for $w$.

3. *Optional*: The vector of importance parameters for the token $p_w$ can be concatenated with $\hat{e}_w$ in order to construct the final hash vector $e_w$.

The full translation of a token to a hash vector can be written in vector notation ($\oplus$ denotes the concatenation operator):

$$
\begin{aligned}
c_w &= (\mathcal{H}_1(w), \mathcal{H}_2(w), \ldots, \mathcal{H}_k(w))^\top \\
p_w &= (p_w^1, \ldots, p_w^k)^\top \\
\hat{e}_w &= p_w^\top c_w \\
e_w^\top &= \hat{e}_w^\top \oplus p_w^\top \text{(optional)}
\end{aligned}
$$

The token to component vector functions $\mathcal{H}_i$ are implemented by $\mathcal{H}_i(w) = E_{D_2(D_1(w))}$, where

- $D_1 : \mathcal{T} \to \{1, \ldots K\}$ is a token to id function.
- $D_2 : \{1, \ldots, K\} \to \{1, \ldots B\}$ is an id to bucket (hash) function.
- $E$ is a $B \times d$ matrix.

If creating a dictionary beforehand is not a problem, we can use an enumeration (dictionary) of the tokens as $D_1$. If, on the other hand, it is inconvenient (or impossible) to use a dictionary because of the size of $\mathcal{T}$, we can simply use a hash function $D_1 : \mathcal{T} \to \{1, \ldots K\}$.

The importance parameter vectors $p_w$ are represented as rows in a $K \times k$ matrix $P$, and the token to importance vector mapping is implemented by $w \to P_{\hat{D}(w)}$. $\hat{D}(w)$ can be either equal to $D_1$, or we can use a different hash function. In the rest of the article we will use $\hat{D} = D_1$, and leave the case where $\hat{D} \neq D_1$ to future work.

Based on the description above we see that the construction of hash embeddings requires the following:

1. A trainable embedding matrix $E$ of size $B \times d$, where each of the $B$ rows is a component vector of length $d$.

2. A trainable matrix $P$ of importance parameters of size $K \times k$ where each of the $K$ rows is a vector of $k$ scalar importance parameters.

3. $k$ different hash functions $\mathcal{H}_1, \ldots, \mathcal{H}_k$ that each uniformly assigns one of the $B$ component vectors to each token $w \in \mathcal{T}$.

The total number of trainable parameters in a hash embedding is thus equal to $B \cdot d + K \cdot k$, which should be compared to a standard embedding where the number of trainable parameters is $K \cdot d$. The number of hash functions $k$ and buckets $B$ can typically be chosen quite small without degrading performance, and this is what can give a huge reduction in the number of parameters (we typically use $k = 2$ and choose $K$ and $B$ s.t. $K > 10 \cdot B$).

From the description above we also see that the computational overhead of using hash embeddings instead of standard embeddings is just a matrix multiplication of a $1 \times k$ matrix (importance parameters) with a $k \times d$ matrix (component vectors). When using small values of $k$, the computational overhead is therefore negligible. In our experiments, hash embeddings were actually marginally faster to train than standard embedding types for large vocabulary problems[1]. However, since the embedding layer is responsible for only a negligible fraction of the computational complexity of most models, using hash embeddings instead of regular embeddings should not make any difference for most models. Furthermore, when using hash embeddings it is not necessary to create a dictionary before training nor to perform vocabulary pruning after training. This can also reduce the total training time.

Note that in the special case where the number of hash functions is $k = 1$, and all importance parameters are fixed to $p_w^1 = 1$ for all tokens $w \in \mathcal{T}$, hash embeddings are equivalent to using the hashing trick. If furthermore the number of component vectors is set to $B = |\mathcal{T}|$ and the hash function $h_1(w)$ is the identity function, hash embeddings are equivalent to standard embeddings.

## 4 Hashing theory

**Theorem 4.1.** *Let $h : \mathcal{T} \to \{0, \ldots, K\}$ be a hash function. Then the probability $p_{col}$ that $w_0 \in \mathcal{T}$ collides with one or more other tokens is given by*

$$p_{col} = 1 - (1 - 1/K)^{|\mathcal{T}|-1} \, . \tag{1}$$

*For large $K$ we have the approximation*

$$p_{col} \approx 1 - e^{-\frac{|\mathcal{T}|}{K}} \, . \tag{2}$$

*The expected number of tokens in collision $C_{tot}$ is given by*

$$C_{tot} = |\mathcal{T}| p_{col} \, . \tag{3}$$

*Proof.* This is a simple variation of the "birthday problem".

$\square$

When using hashing for dimensionality reduction, collisions are unavoidable, which is the main disadvantage for feature hashing. This is counteracted by hash embeddings in two ways:

First of all, for choosing the component vectors for a token $w \in \mathcal{T}$, hash embeddings use $k$ independent uniform hash functions $h_i : \mathcal{T} \to \{1, \ldots, B\}$ for $i = 1, \ldots, k$. The combination of multiple hash functions approximates a single hash function with much larger range $h : \mathcal{T} \to \{1, \ldots, B^k\}$, which drastically reduces the risk of total collisions. With a vocabulary of $|\mathcal{T}| = 100M$, $B = 1M$ different component vectors and just $k = 2$ instead of 1, the chance of a given token colliding with at least one other token in the vocabulary is reduced from approximately $1 - \exp\left(-10^8/10^6\right) \approx 1$ to approximately $1 - \exp\left(-10^8/10^{12}\right) \approx 0.0001$. Using more hash functions will further reduce the number of collisions.

Second, only a small number of the tokens in the vocabulary are usually important for the task at hand. The purpose of the importance parameters is to implicitly prune unimportant words by setting their importance parameters close to 0. This would reduce the expected number of collisions to $|\mathcal{T}_{\text{imp}}| \cdot \exp\left(-\frac{|\mathcal{T}_{\text{imp}}|}{B}\right)$ where $\mathcal{T}_{\text{imp}} \subset \mathcal{T}$ is the set of important words for the given task. The weighting with the component vector will further be able to separate the colliding tokens in the $k$ dimensional subspace spanned by their $k$ $d$ dimensional embedding vectors.

Note that hash embeddings consist of two layers of hashing. In the first layer each token is simply translated to an integer in $\{1, \ldots, K\}$ by a dictionary or a hash function $D_1$. If $D_1$ is a dictionary, there will of course not be any collisions in the first layer. If $D_1$ is a random hash function then the expected number of tokens in collision will be given by equation 3. These collisions cannot be avoided, and the expected number of collisions can only be decreased by increasing $K$. Increasing the vocabulary size by 1 introduces $d$ parameters in standard embeddings and only $k$ in hash embeddings. The typical $d$ ranges from 10 to 300, and $k$ is in the range 1-3. This means that even when the embedding size is kept small, the parameter savings can be huge. In (Joulin et al., 2016b) for example, the embedding size is chosen to be as small as 10. In order to go from a bi-gram model to a general $n$-gram model the number of buckets is increased from $K = 10^7$ to $K = 10^8$. This increase of buckets requires an additional 900 million parameters when using standard embeddings, but less than 200 million when using hash embeddings with the default of $k = 2$ hash functions. I.e. even when the embedding size is kept extremely small, the parameter savings can be huge.

## 5 Experiments

We benchmark hash embeddings with and without dictionaries on text classification tasks.

### 5.1 Data and preprocessing

We evaluate hash embeddings on 7 different datasets in the form introduced by Zhang et al. (2015) for various text classification tasks including topic classification, sentiment analysis, and news categorization. All of the datasets are balanced so the samples are distributed evenly among the

classes. An overview of the datasets can be seen in table 1. Significant previous results are listed in table 2. We use the same experimental protocol as in (Zhang et al., 2015).

We do not perform any preprocessing besides removing punctuation. The models are trained on snippets of text that are created by first converting each text to a sequence of $n$-grams, and from this list a training sample is created by randomly selecting between 4 and 100 consecutive $n$-grams as input. This may be seen as input drop-out and helps the model avoid overfitting. When testing we use the entire document as input. The snippet/document-level embedding is obtained by simply adding up the word-level embeddings.

Table 1: Datasets used in the experiments, See (Zhang et al., 2015) for a complete description.

|  | #Train | #Test | #Classes | Task |
| --- | --- | --- | --- | --- |
| AG's news | 120k | 7.6k | 4 | English news categorization |
| DBPedia | 450k | 70k | 14 | Ontology classification |
| Yelp Review Polarity | 560k | 38k | 2 | Sentiment analysis |
| Yelp Review Full | 560k | 50k | 5 | Sentiment analysis |
| Yahoo! Answers | 650k | 60k | 10 | Topic classification |
| Amazon Review Full | 3000k | 650k | 5 | Sentiment analysis |
| Amazon Review Polarity | 3600k | 400k | 2 | Sentiment analysis |

## 5.2 Training

All the models are trained by minimizing the cross entropy using the stochastic gradient descent-based *Adam* method (Kingma and Ba, 2014) with a learning rate set to $\alpha = 0.001$. We use early stopping with a patience of 10, and use $5\%$ of the training data as validation data. All models were implemented using Keras with TensorFlow backend. The training was performed on a Nvidia GeForce GTX TITAN X with 12 GB of memory.

## 5.3 Hash embeddings without a dictionary

In this experiment we compare the use of a standard hashing trick embedding with a hash embedding. The hash embeddings use $K = 10$M different importance parameter vectors, $k = 2$ hash functions, and $B = 1$M component vectors of dimension $d = 20$. This adds up to 40M parameters for the hash embeddings. For the standard hashing trick embeddings, we use an architecture almost identical to the one used in (Joulin et al., 2016b). As in (Joulin et al., 2016b) we only consider bi-grams. We use one layer of hashing with 10M buckets and an embeddings size of 20. This requires 200M parameters. The document-level embedding input is passed through a single fully connected layer with softmax activation.

The performance of the model when using each of the two embedding types can be seen in the left side of table 2. We see that even though hash embeddings require 5 times less parameters compared to standard embeddings, they perform at least as well as standard embeddings across all of the datasets, except for DBPedia where standard embeddings perform a tiny bit better.

## 5.4 Hash embeddings using a dictionary

In this experiment we limit the vocabulary to the 1M most frequent $n$-grams for $n < 10$. Most of the tokens are uni-grams and bi-grams, but also many tokens of higher order are present in the vocabulary. We use embedding vectors of size $d = 200$. The hash embeddings use $k = 2$ hash functions and the bucket size $B$ is chosen by cross-validation among [500, 10K, 50K, 100K, 150K]. The maximum number of words for the standard embeddings is chosen by cross-validation among [10K, 25K, 50K, 300K, 500K, 1M]. We use a more complex architecture than in the experiment above, consisting of an embedding layer (standard or hash) followed by three dense layers with 1000 hidden units and ReLU activations, ending in a softmax layer. We use batch normalization (Ioffe and Szegedy, 2015) as regularization between all of the layers.

The parameter savings for this problem are not as great as in the experiment without a dictionary, but the hash embeddings still use 3 times less parameters on average compared to a standard embedding.

As can be seen in table 2 the more complex models actually achieve a *worse* result than the simple model described above. This could be caused by either an insufficient number of words in the vocabulary or by overfitting. Note however, that the two models have access to the same vocabulary, and the vocabulary can therefore only explain the general drop in performance, not the performance difference between the two types of embedding. This seems to suggest that using hash embeddings have a regularizing effect on performance.

When using a dictionary in the first layer of hashing, each vector of importance parameters will correspond directly to a unique phrase. In table 4 we see the phrases corresponding to the largest/smallest (absolute) importance values. As we would expect, large absolute values of the importance parameters correspond to important phrases. Also note that some of the $n$-grams contain information that e.g. the bi-gram model above would not be able to capture. For example, the bi-gram model would not be able to tell whether 4 or 5 stars had been given on behalf of the sentence *"I gave it 4 stars instead of 5 stars"*, but the general $n$-gram model would.

## 5.5 Ensemble of hash embeddings

The number of buckets for a hash embedding can be chosen quite small without severely affecting performance. $B = 500 - 10.000$ buckets is typically sufficient in order to obtain a performance almost at par with the best results. In the experiments using a dictionary only about 3M parameters are required in the layers on top of the embedding, while $kK + Bd = 2M + B \times 200$ are required in the embedding itself. This means that we can choose to train an ensemble of models with small bucket sizes instead of a large model, while at the same time use the same amount of parameters (and the same training time since models can be trained in parallel). Using an ensemble is particularly useful for hash embeddings: even though collisions are handled effectively by the word importance parameters, there is still a possibility that a few of the important words have to use suboptimal embedding vectors. When using several models in an ensemble this can more or less be avoided since different hash functions can be chosen for each hash embedding in the ensemble.

We use an ensemble consisting of 10 models and combine the models using soft voting. Each model use $B = 50.000$ and $d = 200$. The architecture is the same as in the previous section except that models with one to three hidden layers are used instead of just ten models with three hidden layers. This was done in order to diversify the models. The total number of parameters in the ensemble is approximately 150M. This should be compared to both the standard embedding model in section 5.3 and the standard embedding model in section 5.4 (when using the full vocabulary), both of which require $\approx 200M$ parameters.

Table 2: Test accuracy (in %) for the selected datasets

|  | Without dictionary | | With dictionary | | |
|  | Shallow network (section 5.3) | | Deep network (section 5.4) | | |
|  | Hash emb. | Std emb | Hash emb. | Std. emb. | Ensemble |
|---|---|---|---|---|---|
| AG | **92.4** | 92.0 | 91.5 | 91.7 | 92.0 |
| Amazon full | 60.0 | 58.3 | 59.4 | 58.5 | **60.5** |
| Dbpedia | 98.5 | 98.6 | 98.7 | 98.6 | **98.8** |
| Yahoo | 72.3 | 72.3 | 71.3 | 65.8 | **72.9** |
| Yelp full | **63.8** | 62.6 | 62.6 | 61.4 | 62.9 |
| Amazon pol | 94.4 | 94.2 | 94.7 | 93.6 | **94.7** |
| Yelp pol | **95.9** | 95.5 | 95.8 | 95.0 | 95.7 |

# 6 Future Work

Hash embeddings are complementary to other state-of-the-art methods as it addresses the problem of large vocabularies. An attractive possibility is to use hash-embeddings to create a word-level embedding to be used in a context sensitive model such as wordCNN.

As noted in section 3, we have used the same token to id function $D_1$ for both the component vectors and the importance parameters. This means that words that hash to the same bucket in the first layer get both identical component vectors and importance parameters. This effectively means that those words become indistinguishable to the model. If we instead use a different token to id function $\hat{D}$ for

Table 3: State-of-the-art test accuracy in %. The table is split between BOW embedding approaches (bottom) and more complex rnn/cnn approaches (top). The best result in each category for each dataset is bolded.

|  | AG | DBP | Yelp P | Yelp F | Yah A | Amz F | Amz P |
|---|---|---|---|---|---|---|---|
| char-CNN (Zhang et al., 2015) | 87.2 | 98.3 | 94.7 | 62.0 | 71.2 | 59.5 | 94.5 |
| char-CRNN (Xiao and Cho, 2016) | 91.4 | 98.6 | 94.5 | 61.8 | 71.7 | 59.2 | 94.1 |
| VDCNN (Conneau et al., 2016) | 91.3 | 98.7 | 95.7 | **64.7** | 73.4 | 63.0 | 95.7 |
| wordCNN (Johnson and Zhang, 2016) | **93.4** | 99.2 | **97.1** | 67.6 | **75.2** | **63.8** | **96.2** |
| Discr. LSTM (Yogatama et al., 2017) | 92.1 | 98.7 | 92.6 | 59.6 | 73.7 | | |
| Virt. adv. net. (Miyato et al., 2016) | | **99.2** | | | | | |
| fastText (Joulin et al., 2016b) | **92.5** | 98.6 | 95.7 | **63.9** | 72.3 | 60.2 | 94.6 |
| BoW (Zhang et al., 2015) | 88.8 | 96.6 | 92.2 | 58.0 | 68.9 | 54.6 | 90.4 |
| $n$-grams (Zhang et al., 2015) | 92.0 | 98.6 | 95.6 | 56.3 | 68.5 | 54.3 | 92.0 |
| $n$-grams TFIDF (Zhang et al., 2015) | 92.4 | 98.7 | 95.4 | 54.8 | 68.5 | 52.4 | 91.5 |
| Hash embeddings (no dict.) | 92.4 | 98.5 | **95.9** | 63.8 | 72.3 | 60.0 | 94.4 |
| Hash embeddings (dict.) | 91.5 | 98.7 | 95.8 | 62.5 | 71.9 | 59.4 | **94.7** |
| Hash embeddings (dict., ensemble) | 92.0 | **98.8** | 95.7 | 62.9 | **72.9** | **60.5** | **94.7** |

Table 4: Words in the vocabulary with the highest/lowest importance parameters.

|  | Yelp polarity | Amazon full |
|---|---|---|
| Important tokens | What_a_joke, not_a_good_experience, Great_experience, wanted_to_love, and_lacking, Awful, by_far_the_worst, | gave_it_4, it_two_stars_because, 4_stars_instead_of_5, 4_stars, four_stars, gave_it_two_stars |
| Unimportant tokens | The_service_was, got_a_cinnamon, 15_you_can, while_touching, and_that_table, style_There_is | that_my_wife_and_I, the_state_I, power_back_on, years_and_though, you_want_a_real_good |

the importance parameters, we severely reduce the chance of "total collisions". Our initial findings indicate that using a different hash function for the index of the importance parameters gives a small but consistent improvement compared to using the same hash function.

In this article we have represented word vector using a weighed sum of component vectors. However, other aggregation methods are possible. One such method is simply to concatenate the (weighed) component vectors. The resulting $kd$-dimensional vector is then equivalent to a weighed sum of orthogonal vectors in $\mathbb{R}^{kd}$.

Finally, it might be interesting to experiment with pre-training lean, high-quality hash vectors that could be distributed as an alternative to word2vec vectors, which require around 3.5 GB of space for almost a billion parameters.

# 7 Conclusion

We have described an extension and improvement to standard word embeddings and made an empirical comparisons between hash embeddings and standard embeddings across a wide range of classification tasks. Our experiments show that the performance of hash embeddings is always at par with using standard embeddings, and in most cases better.

We have shown that hash embeddings can easily deal with huge vocabularies, and we have shown that hash embeddings can be used both with and without a dictionary. This is particularly useful for problems such as online learning where a dictionary cannot be constructed before training.

Our experiments also suggest that hash embeddings have an inherent regularizing effect on performance. When using a standard method of regularization (such as $L_1$ or $L_2$ regularization), we start with the full parameter space and regularize parameters by pushing some of them closer to 0. This is in contrast to regularization using hash embeddings where the number of parameters (number of buckets) determines the degree of regularization. Thus parameters not needed by the model will not have to be added in the first place.

The hash embedding models used in this article achieve equal or better performance than previous bag-of-words models using standard embeddings. Furthermore, in 5 of 7 datasets, the performance of hash embeddings is in top 3 of state-of-the art.

## Footnotes

[1] the small performance difference was observed when using Keras with a Tensorflow backend on a GeForce GTX TITAN X with 12 GB of memory and a Nvidia GeForce GTX 660 with 2GB memory. The performance penalty when using standard embeddings for large vocabulary problems can possibly be avoided by using a custom embedding layer, but we have not pursued this further.

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
