[Reviews · NeurIPS 2017]

Reviewer 1



This work describes an extension of the standard word hashing trick for embedding representation by using a weighted combination of several vectors indexed by different hash functions to represent each word. This can be done using a predefined dictionary or during online training. The approach has the benefit of being easy to understand and implement and greatly reduces the number of embedding parameters. The results are generally good and the approach is quite elegant. Additionally, the hash embeddings seem to act as an effective regularizer. However, table 2 would benefit from describing the final selected vocabulary sizes as well as the parameter reduction provided by the hash embedding. Table 3 is also missing a joint state of the art model for the DBPedia dataset [1]. Line 255 makes the claim that the ensemble would train in the same amount of time as the single large model. However, this would only be true if each model in the ensemble had an architecture with fewer weights (that were not embedding weights). From the description, it seems that the number of non-embedding weights in each network in the ensemble is the same as that in the large model so that training time would be significantly larger for the ensemble. Table 3 highlights the top 3 best models, however, a clearer comparison might be to split the table into embedding only approaches vs RNN/CNN approaches. It would also be interesting to see these embeddings used in the more context-sensitive RNN/CNN models. Minor comments: L9: Typo: million(s) L39: to(o) much L148: This sentence is awkwardly rewritten. L207: What is patience? L235: Typo: table table Table 4: Were these obtained through summing the importance weights? What were the order of the highest/lowest weights? [1] Miyato, Takeru, Andrew M. Dai, and Ian Goodfellow. "Virtual adversarial training for semi-supervised text classification." ICLR 2017.

Reviewer 2



This paper uses hashed embeddings to reduce the memory footprint of the embedding parameters. Their approach is simple, where they learn a vector of importance parameters, one for each component vectors. Both the trainable matrices: the matrix of importance parameters and tThe embedding matrix of component vectors is much smaller than a regular embedding matrix. To obtain an embedding for a word, they first hash the token id into row of the importance parameters and then hash each component of the importance parameters into the component vector embedding. Their results show that just using bow models, their ensemble of hash embeddings do better than previous bow models on classification tasks. This is a simple yet effective approach to decrease the number of parameters in the model and can also serve as a regularizer. The authors should point out the compute time required for inference and if hashing is more expensive than an embedding gather on the GPU.

Reviewer 3



This paper proposes hash embeddings to obtain vector representations of words using hashing tricks. This model uses a set of hash functions to map a word to a small set of shared embedding vectors. This technique enables to reduce the cumulative number of parameters and to avoid difficulties the conventional word embedding model have. This model is simple, easy to implement, and achieves state-of-the-art experimental results. First of all, the question I have is about the difficulty of training hash embeddings. This model has to train both a set importance weights on the hash function and shared embeddings. In my intuition, it makes the training difficult to obtain a reasonable local minimum because these two parameters affect each other. If possible, it is better to show how robust this model is regarding initialization points and hyper-parameters, and give a good intuition about how to find a good local minimum. The second question is the relationship between "discourse atoms" [1] and shared embedding trained from hash embeddings. "discourse atoms" also try to reconstruct word embeddings by a linear combination of a small set of atom vectors. As a result, some atoms could capture higher concepts about words (like ontology), and I guess these characteristics can also be captured in shared embeddings in hash embeddings. Do the authors have some insights about the relationship between these "atoms" and hashed representations. [1] Linear Algebraic Structure of Word Senses, with Applications to Polysemy https://arxiv.org/abs/1601.03764